# Isolation, Identification and Hyperparasitism of a Novel *Cladosporium cladosporioides* Isolate Hyperparasitic to *Puccinia striiformis* f. sp. *tritici*, the Wheat Stripe Rust Pathogen

**DOI:** 10.3390/biology11060892

**Published:** 2022-06-10

**Authors:** Hongjuan Zhang, Mengying He, Xin Fan, Lu Dai, Shan Zhang, Zeyu Hu, Ning Wang

**Affiliations:** 1Department of Biological Engineering, Yangling Vocational & Technical College, Xianyang 712100, China; zhanghongjuancn@126.com (H.Z.); dailuxn@126.com (L.D.); 2State Key Laboratory of Crop Stress Biology for Arid Areas, College of Plant Protection, Northwest A&F University, Xianyang 712100, China; 18392053924@163.com (M.H.); m15029926806_1@163.com (X.F.); dorothyz1997@163.com (S.Z.); huzeyu06@126.com (Z.H.)

**Keywords:** *Puccinia striiformis* f. sp. *tritici* (*Pst*), *Cladosporium cladosporioides*, hyperparasite, microbial biological control agents (MBCAs)

## Abstract

**Simple Summary:**

Obligate biotrophic pathogen *Puccinia striiformis* f. sp. *tritici* (*Pst*) is a major threat to wheat production. Parasites of *Pst* can be used to develop biological agents for environmentally friendly control of this fungal disease. Here, we report a hyperparasitic fungus isolated from taupe-colored uredinia of *Pst* and identified as *Cladosporium cladosporioides* through molecular and morphological characterizations. The hyperparasitic isolate was able to reduce the production and viability of *Pst*. Therefore, *Cladosporium cladosporioides* may have potential in biological control of stripe rust on wheat.

**Abstract:**

Wheat rust outbreaks have caused significantly economic losses all over the world. *Puccinia striiformis* f. sp. *tritici* (*Pst*) is an obligate biotrophic fungus causing stripe rust on wheat. Application of fungicides may cause environmental problems. The effects of hyperparasites on plant pathogens are the basis for biological control of plant pathogenic fungi and parasites of *Pst* have great value in biological agents development. Here, we report the isolation and characterization of isolate of *Cladosporium cladosporioides* from *Pst* based on morphological characterization and analysis of molecular markers. The hyperparasitic isolate was isolated from taupe-colored uredinia of *Pst*. Upon artificial inoculation, the hyperparasitic isolate was able to reduce the production and germination rate of *Pst* urediospores, and *Pst* uredinia changed color from yellow to taupe. Scanning electron microscopy demonstrated that the strain could efficiently colonize *Pst* urediospores. Therefore, the isolate has the potential to be developed into a biological control agent for managing wheat stripe rust.

## 1. Introduction

Wheat stripe rust (also called yellow rust), caused by *Puccinia striiformis* f. sp. *tritici* (*Pst*), poses a great threat to wheat production worldwide [1]. In 2000, 9 of the 64 major wheat producing countries reported severe losses in wheat yields caused by stripe rust [2]. In 2000–2012, about 88% of the world’s wheat-producing areas were affected by stripe rust [3]. In China, the disease can reduce the yield of wheat by 10–20%, and even more than 60% in extremely severe epidemic years [4,5].

Wheat stripe rust is currently controlled mainly by developing resistant cultivars and applying pesticides when needed. However, race-specific resistance is often circumvented by new races of the pathogen, and pesticides application adds extra cost and may adversely affect the environment [6,7,8]. Therefore, other strategies such as biological control could provide additional options for managing stripe rust. Hyperparasitic fungi have been reported for various plant pathogenic fungi [9,10,11], and some of them have been used in control of plant diseases [11,12,13]. *Ampelomyces* mycoparasites suppressed the sporulation rate of powdery mildew, and the infected plants regained vigor after *Ampelomyces* killed the pathogens [14]. The cell wall protein fraction of *Pythium oligandrum*, a parasite of pathogen *Rhizoctonia solani* AG-3, enhanced the expression of defense-related genes in potato. Application of *P. oligandrum* oospore suspension at 10^4^ or 10^5^ mL^−1^ to potato seed tuber pieces, the black scurf disease severity index reduced about 35% compared with untreated controls in field [15]. More than 30 genera of hyperparasitic fungi parasitic in rust fungi have been reported, including *Alternaria* spp., *Aphanocladium* spp., *Fusarium* spp., *Lecanicillium* spp., and *Scytalidium* spp. [16,17,18,19,20,21]. However, only six species, *Alternaria alternata*, *Cladosporium cladosporioides*, *Lecanicillium Lecanii*, *Microdochium nivale*, *Simplicillium obclavatum* and *Typhula idahoensis* have been reported to infect uredinia and urediospores of *Pst* [18,21,22,23].

*Pst* is an obligate biotrophic basidiomycete fungus. The fungus produces yellow to orange uredinia on susceptible host plants. During an *Pst* urediospores inoculation experiment in a growth chamber, we found that yellow to orange uredinia turned taupe. Based on this phenomenon, we further isolated a novel *Pst* hyperparasite. Based on morphological characterization and analysis of molecular marker, we identified the hyperparasitic isolate as *C. cladosporioides* (Fresen.) G.A. de Vries. Furthermore, we demonstrated that this isolate was able to reduce the production and viability of *Pst* urediospores. Thus, the isolate may have potential in biological control of wheat stripe rust.

## 2. Materials and Methods

### 2.1. Isolation of the Hyperparasite from Pst-Infected Leaves

Wheat cultivar “Fielder” inoculated with *Pst* urediospores were kept in a growth chamber at about 16 °C and 80–90% relative humidity. When *Pst* was sporulating 14 days after inoculation, *Pst* uredinia started to change color from yellow to taupe. Leaves bearing taupe pustules were cut off the plants, surface-sterilized with 75% alcohol for 1 min, and transferred to a Petri dish containing PDA medium. The dish was incubated in darkness at 25 °C for 5 days [21]. A mycelial tip was transferred to a new dish and incubated under the same condition for obtaining a pure culture.

### 2.2. Morphological Identification

The obtained pure culture isolate, *C. cladosporioides* R23Bo, was grown on PDA as described above, and a diameter of 5 mm mycelial disk was placed at the center of a new PDA plate and cultured at the same conditions. Colonies, hyphae, conidiophores, and conidia were observed and measured under a light microscope.

To study the isolate’s ultrastructure, the samples were prepared using the previously described method [21]. First, the samples were fixed in a glutaraldehyde fixative solution overnight at 4℃, rinsed with PBS buffer for 10 min for 4 times, and dehydrated for 15–20 min with five concentration gradients (30%, 50%, 70%, 80%, 90%) ethanol. The dehydrated samples were soaked in isoamyl acetate for 10–20 min, and then processed with carbon dioxide drier. Finally, the samples were treated by spray-gold [24]. The samples were observed under a SEM.

### 2.3. Molecular Characterization

Mycelia of isolate R23Bo were collected from colonies cultured at 25 °C in darkness for 5 days. DNA was extracted from the mycelia using the cetyl trimethylammonium bromide (CTAB) method [25]. The generic primers of ITS (eukaryotic ribosomal DNA) (ITS1: TCCGTAGGTGAACCTGCG; ITS4: TCCTCCGCTTATTGATATGC) were used in PCR amplification. PCR procedure was conducted as follows: 94 °C for 4 min; 94 °C for 30 s, 55 °C for 30 s, 72 °C for 30 s, 35 cycles; 72 °C for 10 min. The PCR products were separated in 1.5% agarose gel and collected and purified using the agarose gel DNA extraction purification Kit (Takara, Dalian, China). The amplified fragments were sequenced by the AuGCT company (Beijing, China).

### 2.4. Phylogenetic Analysis

The sequences of five species in genus *Cladosporium* were retrieved from GenBank (Table 1), and aligned using software MEGA7.0.26 (https://www.megasoftware.net/ (accessed on 5 May 2022)) [26]. Phylogenetic analysis was conducted using the neighbor-joining (NJ) method, and bootstrap analysis was conducted to determine the robustness of branches using 1000 replications.

### 2.5. Pathogenicity and Hyperparasite Tests

The inoculations of *Pst* were performed following the previously described methods [21]. Briefly, wheat plants (cv. Fielder) grown in a greenhouse for 20 days were first inoculated with urediospores of *Pst* race CYR31 collected from Su11 wheat. The collected urediospores of *Pst* race CYR31 were diluted with water to 20 mg·mL^−1^ and inoculated by brush. The *Pst*-inoculated plants were incubated in a dew chamber at 12 °C in dark for 24 h, and then grown in a growth chamber at 16 °C with 16 h light photoperiod. Three, five, seven and nine days after *Pst* inoculation, the plants in different pots were inoculated with the conidian suspension (1.0 × 10^6^ spores/mL) of *C. cladosporioides* isolate R23Bo, kept in a dew chamber at 16 °C in dark for 24 h, and then returned to the growth chamber for growth under the same conditions. Plants inoculated only with *Pst* urediospores were used as a control. Fourteen days after *Pst* inoculation, symptoms and signs were recorded and yellow colored uredinia were counted using pictures analyzed with Image J number counting software (National Institutes of Health, 1.48u, Bethesda, MD, USA). Samples for SEM observation were collected at 3, 5, 7, and 9 days after hyperparasite inoculation (dai). Microscopic observations were conducted using a SEM. Genomic DNA of the samples were extracted from the infected leaf tissue to determine the fungi biomass (*Pst* DNA/wheat DNA ratio) at 3, 5, 7, 9 dai. Quantitative PCR (qPCR) was performed in a CFX96 Connect Real-Time PCR Detection System (Bio-Rad, Hercules, CA, USA) to determine the *Pst* DNA content in the infected wheat leaves using a TB Green Premix DimerEraser (Perfect Real Time) (TaKaRa, Dalian, China). The fungal *Pst-EF1* (*Pst* elongation factor 1) and wheat *TaEF1-α* (wheat elongation factor 1 alpha) fusion plasmids were diluted into a serial concentrations (10^3^, 10^4^, 10^5^, 10^6^, 10^7^, 10^8^, and 10^9^ fmol·cotL^−1^) for generation of the standard curves (Appendix A). Wheat-*EF1* primers (F: TGGTGTCATCAAGCCTGGTATGGT; R: ACTCATGGTGCATCTCAACGGACT) and *Pst-EF1* primers (F: TTCGCCGTCCGTGATATGAGACAA; R: ATGCGTATCATGGTGGTGGAGTGA) were used for qPCR analysis. The experiment include three independently biological repeats. Germination rate of *Pst* assay was performed following the previously described methods [21] with some modifications. Freshly collected hyperparasited urediospores were cultured on sterile water at 9 °C for 6 h, then placed on slides to count the numbers of germinated urediospores using an Olympus BX51T-32P01 optical microscope (Tokyo, Japan). A germ tube length up to the one-half spore diameter was defined as germination. The germination rate was calculated as germinated urediospores/100 urediospores. One hundred urediospores were selected randomly, and all experiments were performed three times.

## 3. Results

### 3.1. SEM Observations of Pst Uredinia Parasitized by the Hyperparasite

Observations using a scanning electronic microscope (SEM) showed that without hyperparasite infection, *Pst* uredinia had a normal shape and structure (Figure 1A). The *Pst* urediospores were shriveled at the early infection stage of the hyperparasite (Figure 1B–D). Soon, the hyperparasite hyphae invaded the urediospores (Figure 1E,F), and the urediospores were completely covered by hyperparasite conidian and hyphae and eventually disappeared (Figure 1G,H).

### 3.2. Morphological Characterization of the Hyperparasite

The morphological characteristics of the hyperparasite cultured on media were studied using a light microscope. The front side of the fungal colonies was taupe-colored and the reverse side was brown to black (Figure 2A,B). Mycelia grew fast and were dense. Colonies reached 20–30 mm in diameter on potato dextrose agar (PDA), at 25 °C in 7 days (Figure 2A). The conidiophores were light brown and branched, and conidia varied in size, ranging from 5 µm to 15 µm (mean 12 µm) in length and from 1 µm to 5 µm (mean 2.5 µm) in width (Figure 2C,D).

More detailed morphological features of the hyperparasite were revealed by SEM observations. Ramoconidia had one or more conidial scars (Figure 3A–D). Numerous conidia eventually form clusters (Figure 3E–H). According to the above morphology characteristics, the hyperparasite was identified as *Cladosporium cladosporioides* (Fresen.) G.A. de Vries.

### 3.3. Molecular Characterization of the C. cladosporioides Isolate

A neighbor-joining (NJ) tree was constructed for with *Cladosporium* species based on the internal transcribed spacer (ITS) sequences using software MEGA7 (Figure 4). The *Cladosporium* species used for the phylogenetic analysis are provided in Table 1. Our isolate R23Bo was most closely related to isolate QTYC16 of *C. cladosporioides* previously isolated from *Pantala flavescens* larvae, but not closely related to 14PI001, an isolate of *C. cladosporioides* previously isolated from *Pst* (Figure 4) Isolate.

### 3.4. Confirmation of the C. cladosporioides Isolate Parasitizing Pst

The hyperparasitic ability of isolate R23Bo of *C. cladosporioides* was confirmed by co-inoculation of wheat plants with *Pst* and the isolate. Wheat leaves inoculated only with the conidian suspension of isolate R23Bo did not show any symptoms or signs of fungal infection (Figure 5A). When inoculated with only the *Pst* urediospores suspension, yellow colored uredinia with urediospores formed on the inoculated leaves 12 days post inoculation (dpi) (Figure 5B). When wheat leaves were inoculated with *Pst* urediospores followed by inoculation with the conidia suspension of the *C. cladosporioides* isolates different days after *Pst* inoculation, *Pst* yellow-colored uredinia were changed to taupe (Figure 5A,C–F). The longer *C. cladosporioides* grew together with *Pst*, the fewer yellow uredinia or the more taupe pustules.

When the number of yellow uredinia were counted using ImageJ number counting software, 12 days after *Pst* inoculation, the number of yellow uredinia per cm^2^ leaves was the lowest in the treatment of *C. cladosporioides* 9 days after *Pst* inoculation (Figure 6A). R23Bo-strain-treated pustules showed impact the production of urediospores, and the fertility of spores is seriously affected, as is obviously exhibited in that the ratio of the spores germination reduces by 65% at 3 dpi and 80% at 5 dpi (Figure 6B). The biomass of the *Pst*, measured by the *Pst* DNA/wheat DNA ratio, decreased as the treatment with *C. cladosporioides* lengthened (Figure 6C). The results showed that isolate of *C. cladosporioides* is able to parasitize *Pst*, leading to the reduction in *Pst* urediospore production.

SEM observation further illustrated that isolate R23Bo could efficiently parasitize *Pst*. At 36 hai, *C. cladosporioides* conidian produced germ tubes which contacted with *Pst* urediospores and then grew into the urediospores (Figure 7B). The parasitic fungus grew inside and produced hyphae and conidiophores from the urediospores (Figure 7C), and it completely destroyed the urediospores at 120 h after the parasite treatment (Figure 7C,D).

## 4. Discussion

The identification of new hyperparasites is useful to understanding the biodiversity of mycoparasites, and it provides the potential to develop new strategies for biological control of plant diseases [13,32]. In the present study, we isolated and identified a fungal isolate from *Pst* uredinia. The hyperparasitic isolate is able to reduce *Pst* infection. Furthermore, the isolate can reduce *Pst* urediospore production and viability. Thus, the isolate has a potential value in biological prevention of wheat stripe rust.

In the morphological identification, the spore size is an important classification criterion of *Cladosporium* spp. As the spore dimensions of the most species in the genus overlap, it is difficult to identify species of *Cladosporium* using only morphological characters, especially the size of conidian [28]. Using only ITS sequences is also not reliable to identify *Cladosporium* spp. [33]. In the present study, we use ITS sequence analysis and morphological features to identify the hyperparasitic isolate as *C. cladosporioides*. Several fungal species have been reported to parasitize *Pst,* including *C. cladosporioides* [22]. However, the ITS sequence analysis showed that the *C. cladosporioides* isolate obtained in this study is clearly different from the isolate reported in Zhan et al. [22]. It is interesting that our isolate is most closely related to a *C. cladosporioides* isolate obtained from *Pantala flavescens* larvae [29]. This relationship may suggest that the isolate we obtained from *Pst* uredinia may have other hosts to parasitize and/or natural substrates to grow on.

Biocontrol strategies have potential to achieve efficacy in preventing and treating diseases under environmentally friendly conditions. Some studies have been conducted to explore hyperparasites to control rusts. For example, *Cladosporium* spp. was found to parasitize *Melampsora* spp. [34]. Several fungal species including *C. cladosporioides* were identified as hyperparasites of *Pst* [18,21,22]. The isolate R23Bo of *C. cladosporioides* identified in the present study is able to reduce or stop the growth of *Pst* urediospores by growing into uredinia. The isolate is fast-growing and easy to culture. Additionally, this parasitic ability makes R23Bo a potential biological control agent which could be developed into a biocontrol agent for managing wheat stripe rust.

Stripe rust is started by urediospore infection of host plants and continually develops by producing more urediospores and consequently more infections. Therefore, it is crucial to reduce urediospores for combating the rust disease. In the present study, we observed that after the inoculation of *Pst* urediospores and *C. cladosporioides* conidian on wheat leaves, urediospores were first produced on the wheat leaves, and then *C. cladosporioides* began to grow on urediospores. The exact invasion or parasitism stage cannot be determined at present. It is only clear that *C. cladosporioides* parasitizes in the sporulation stage of *Pst*. In order to develop the isolate as a biocontrol agent, further studies should be conducted to its effect on other plants, humans, animals, and environment, as well as to develop methods for producing and applying the biocontrol agent.

## 5. Conclusions

Identification of parasites infecting cereal pathogenic fungi is essential for developing biological control strategies for managing plant diseases. In this study, we report the discovery of a fungal strain isolated from *Pst*. Through molecular and morphological characterizations, we identified the hyperparasitic fungus as species *Cladosporium cladosporioides*. We demonstrated that the fungus was able to parasitize the obligate biotrophic rust fungus. Our experiments showed that *Cladosporium cladosporioides* was able to impair *Pst* sporulation and reduce urediospores germination. Collectively, *Cladosporium cladosporioides* may be harnessed for controlling stripe rust, and these results shed new light on biological control agent for managing plant pathogens.

The present study identified *Cladosporium cladosporioides* as a new hyperparasite of *Pst*. Although the fungus has the potential utility value as a biological control agent for control stripe rust, additional research is needed to determine if the hyperparasite is environmentally friendly and further to explore its potential to control other rust pathogens.

## Figures and Tables

**Figure 1 biology-11-00892-f001:**
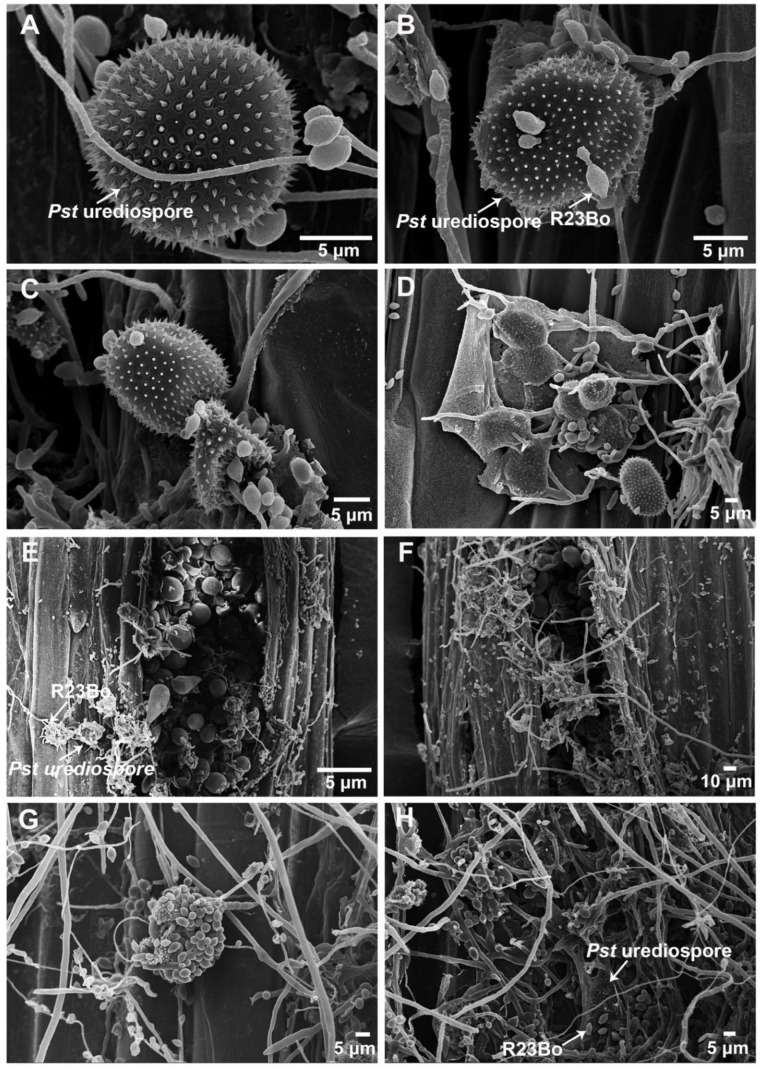
The hyperparasite *Cladosporium cladosporioides* isolate parasitized *Puccinia striiformis* f. sp. *tritici* (*Pst*) urediospores and uredinium. (**A**), urediospores of the *Pst* before *C. cladosporioides* was parasitized (×4000). (**B**–**D**), At the early hyperparasitic stage, the urediospores were shriveled (×3000, ×2200, ×1000, respectively). (**E**,**F**), At the middle hyperparasitic stage, the hyperparasite mycelia and conidia invaded the *Pst* urediospores (×400, ×400, respectively). (**G**,**H**), At the late hyperparasitic stage, the urediospores of the *Pst* are completely destroyed (×1000, ×800, respectively).

**Figure 2 biology-11-00892-f002:**
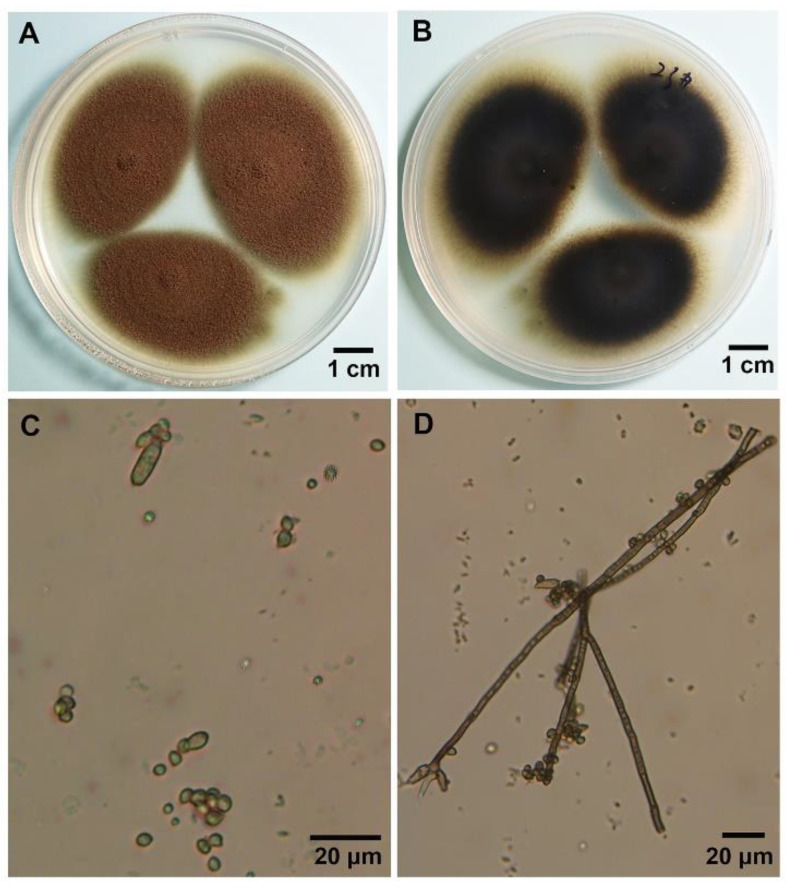
Morphological characteristics of the *Cladosporium cladosporioides* isolate under a light microscope. (**A**,**B**), *C. cladosporioides* cultured on PDA medium at 25 °C for 9 days, the front side and the reverse side, respectively. (**C**,**D**), Septate hyphae and conidia.

**Figure 3 biology-11-00892-f003:**
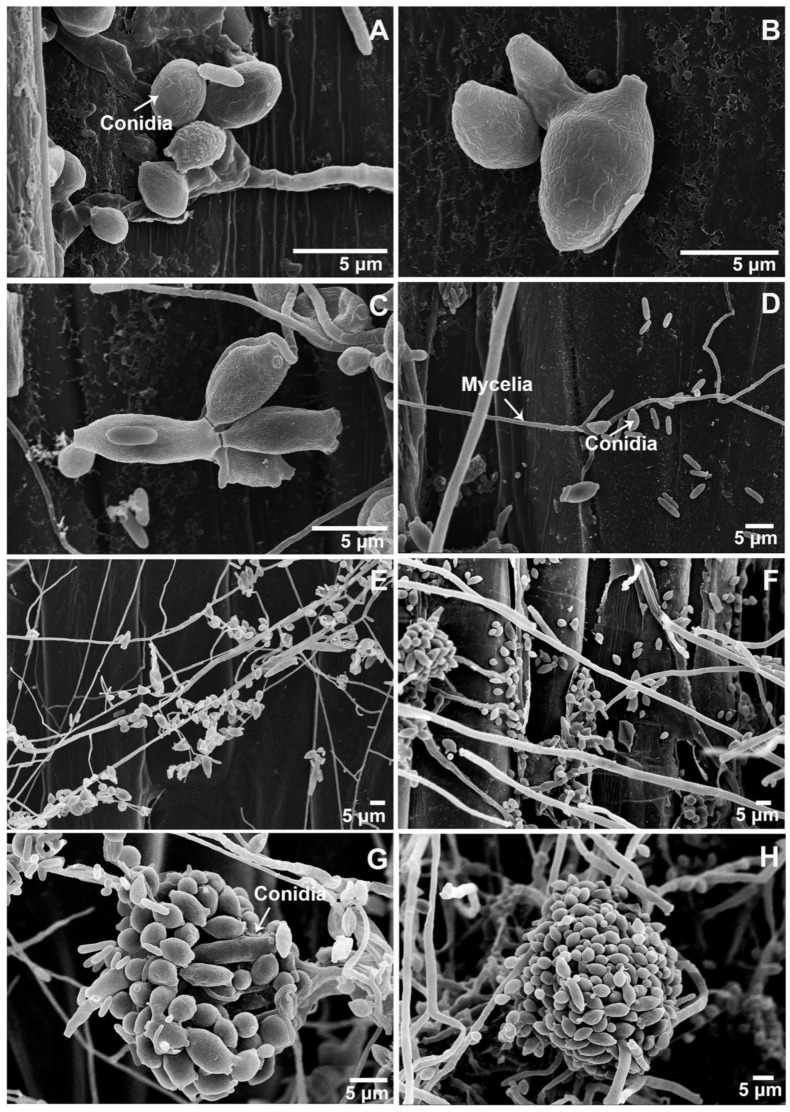
Morphologic characteristics of the *Cladosporium cladosporioides* isolate under a scanning electron microscope. (**A**–**C**), Conidia (×6000, ×13,000 and ×5000, respectively). (**D**–**F**), Mycelia and conidia (×2000, ×1000, and ×1000, respectively). (**G**,**H**), Conidia (×2500 and ×1500, respectively).

**Figure 4 biology-11-00892-f004:**
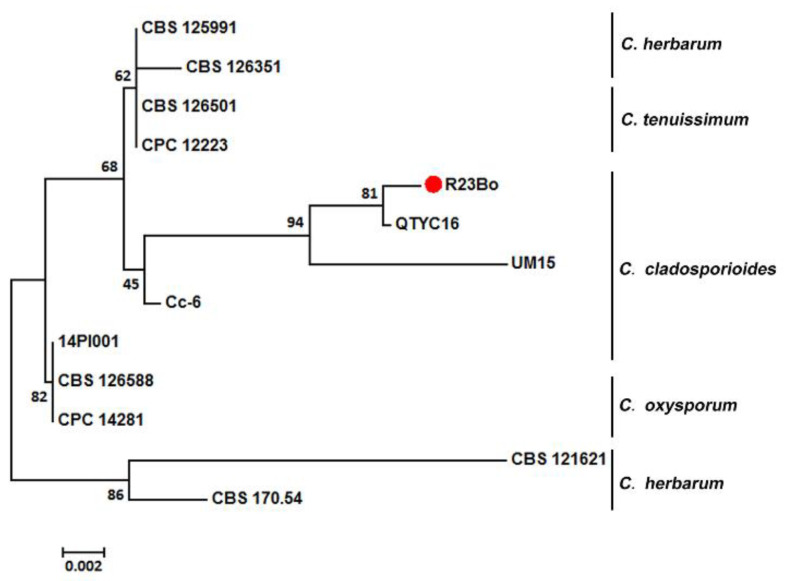
A neighbor-joining (NJ) tree constructed for 13 isolates of 5 *Cladosporium* species using software MEGA7. The *Puccinia striiformis* hyperparasitic isolate (R23Bo) of *C. cladosporioides* is marked with a red dot. The values are the frequencies in percentage for the branches with 1000 bootstrap replications.

**Figure 5 biology-11-00892-f005:**
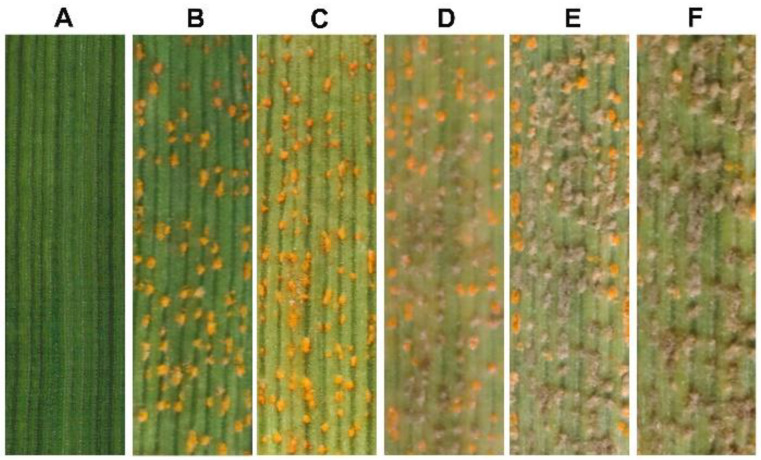
The *Cladosporium cladosporioides* isolate hyperparasitizing *Puccinia striiofrmis* f. sp. *tritici (Pst*). (**A**), Wheat leaf inoculated only with the conidian suspension of *C. cladosporioides* isolate R23Bo; (**B**), At 12 days post inoculation (dpi), wheat leaves inoculated only with *Pst* urediospores; (**C**–**F**), Inoculations with the conidian suspension of *C. cladosporioides* after wheat leaves inoculated with *Pst* (pictures taken 12 dpi of *Pst*). C–F are *Pst*-infected leaves inoculated with *C. cladosporioides* at 3, 5, 7 and 9 dpi of *Pst*, respectively.

**Figure 6 biology-11-00892-f006:**
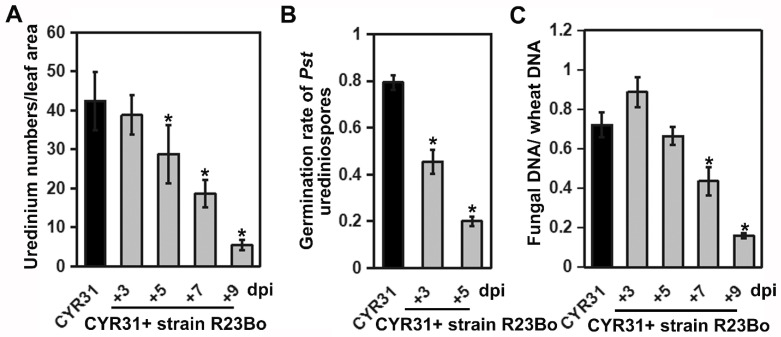
(**A**), Comparison of yellow-colored *Pst* uredinium numbers among the different *Pst* inoculation and time-point treatments of *C. cladosporioides* R23Bo. Yellow-colored uredinia were counted 12 dpi of *Pst* race CYR31. In the legend, CYR31 means only inoculated with CYR31; and CYR31+ 3, 5, 7, and 9 means CYR31 plus inoculation of *C. cladosporioides* R23Bo at 3, 5, 7, and 9 days after the *Pst* inoculation. (**B**), *Pst* urediospores germination rate. (**C**), The *Puccinia striiformis* f. sp. *tritici* biomass changed when hyperparasite *C. cladosporioides* R23Bo parasitized, measured by qRT-PCR. (+) means after *C. cladosporioides* R23Bo treatment 3, 5, 7, and 9 days. Statistical analysis was performed by Student’s *t*-tests. Asterisks indicate *p* value < 0.05.

**Figure 7 biology-11-00892-f007:**
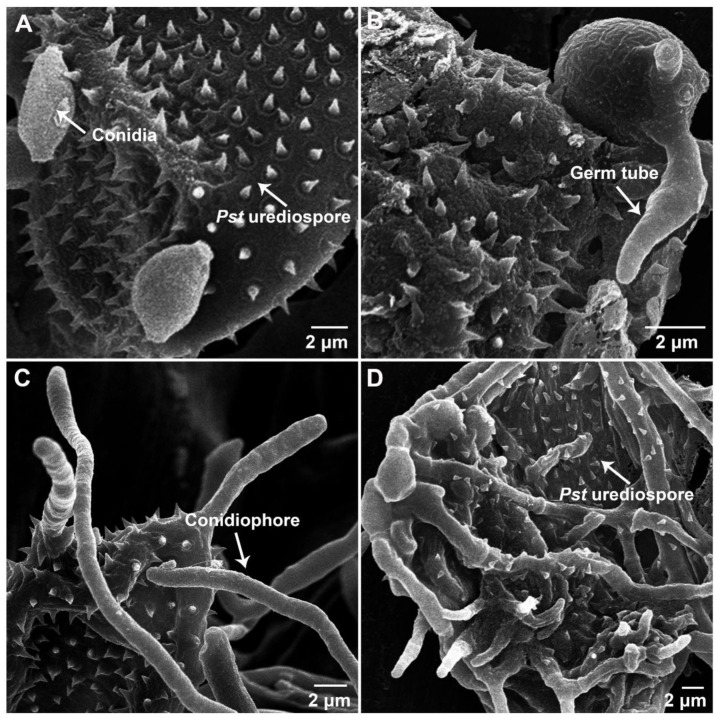
Morphologic characterization of *Puccinia striiformis* f. sp. *tritici* (*Pst*) urediospores infected by *Cladosporium cladosporioides* under a scanning electron microscope. (**A**), Conidia of *C. cladosporioides* on the surface of a *Pst* urediospore (×2000) at 12 hai (hours after inoculation); (**B**), at 36 hai, a *C. cladosporioides* conidium generated a germ tube (×4000); (**C**), at 72 hai, t *C. cladosporioides* produce conidiophores from the *Pst* urediospore (×3000); (**D**), at 120 hai, *C. cladosporioides* has completely colonized the *Pst* urediospore and the urediospore is destroyed (×1500).

**Table 1 biology-11-00892-t001:** Accession and GenBank numbers of *Cladosporium* species used in the molecular characterization of this study.

Species	Accession Number	GenBank Number (ITS, EF, ACT)	Substrate (Including Host)	Country	Reference
*C. cladosporioides*	Cc-6	EU935608.1	*Carica papaya*	China	Chen et al., 2009 [27]
*C. cladosporioides*	UM15	HQ148094.1	*Echinacea purpurea*	USA	Bensch et al., 2010 [28]
*C. cladosporioides*	QTYC16	KM103301.1	*Pantala flavescens* larvae	China	Shao et al., 2015 [29]
*C. cladosporioides*	FCBP:1493	KY290222.1	*Solanum melongena*	Pakistan	-
*C. cladosporioides*	CBS 112388	HM148003 HM148244 HM148490	Indoor air	Germany	Bensch et al., 2010 [28]
*C. cladosporioides*	CBS 143.35	HM148011 HM148252 HM148498	*Pisum sativum*	South Africa	Bensch et al., 2010 [28]
*C. cladosporioides*	CBS 144.35	HM148012 HM148253 HM148499	*P. sativum*	USA	Bensch et al., 2010 [28]
*C. cladosporioides*	CBS 101367	HM148002 HM148489 HM148243	Soil	Brazil	Bensch et al., 2010 [28]
*C. cladosporioides*	CPC 14018 MRC 10810	HM148040 HM148281 HM148527	*Triticum aestivum*	South Africa	Bensch et al., 2010 [28]
*C. cladosporioides*	CPC 14244	HM148044 HM148285 HM148531	*Magnolia* spp.	USA	Bensch et al., 2010 [28]
*C. cladosporioides*	CBS 145.35	HM148013 HM148254 HM148500	*P. sativum*	Germany	Bensch et al., 2010 [28]
*C. cladosporioides*	14PI001 CGMCC 7.175	KJ598781 KM281945 KJ598781	*Puccinia striiformis*	China	Zhan et al., 2014 [22]
*C. herbarum*	CBS 121621	EF679363	*Hordeum vulgare*	Netherlands	Schubert et al., 2007 [30]
*C. herbarum*	CBS 132.29	HM148010 HM148251 HM148497	-^a^	-	Bensch et al., 2010 [28]
*C. herbarum*	CBS 170.54	MH857281 IMI025324 NCTC6740	-	England	Vu et al., 2018 [31]
*C. oxysporum*	CBS 125991 CPC 14371 BA 1738	HM148118 HM148362 HM148607	Soil, near the terracotta	China	Bensch et al., 2010 [28]
*C. oxysporum*	CBS 126351 CPC 14308 BA 1707	HM148119 HM148363 HM148608	Indoor air	Venezuela	Bensch et al., 2010 [28]
*C. pseudocladosporioides*	CBS 149.66	HM148161 HM148405 HM148650	*T. aestivum*	USA	Bensch et al., 2010 [28]
*C. pseudocladosporioides*	CBS 176.82	HM148162 HM148406 HM148651	*Pteridium aquilinum*	Romania	Bensch et al., 2010 [28]
*C. salinae*	CBS 119413 EXF-335	DQ780374 JN906993 EF101390	*Hypersaline water*	Slovenia	Bensch et al., 2010 [28]
*C. tenuissimum*	CPC 12223	HM148208 HM148453 HM148698	Rust	Brazil	Bensch et al., 2010 [28]
*C. tenuissimum*	CBS 126501 CPC 14410	HM148199 HM148444 HM148689	*Musa* spp.	Ivory Coast	Bensch et al., 2010 [28]
*C. xylophilum*	CBS 126588	HM148231 HM148477 HM148722	*Salix viminalis* twigs	Italy	Bensch et al., 2010 [28]
*C. xylophilum*	CPC 14281	HM148233 HM148479 HM148724	Leaves	France	Bensch et al., 2010 [28]

^a^ Not available.

## Data Availability

The relevant datasets supporting the results of this article are included within the article and its additional files.

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
