# Peer review of "Isolation, Identification and Hyperparasitism of a Novel *Cladosporium cladosporioides* Isolate Hyperparasitic to *Puccinia striiformis* f. sp. *tritici*, the Wheat Stripe Rust Pathogen"

_biology, 2022, doi:10.3390/biology11060892_

Round 1

Reviewer 1 Report

Overall, the paper is interesting and was consistently improved. A few short suggestions for authors:

Introduction

Line 36:  10–20%

Lines 42-43: Hyperparasitic fungi have been reported for various plant pathogenic fungi [9–11], and some of them have been used to control plant diseases [11–13].

Materials and methods

Consider not using the 1st person when describing your work (for example, we found can be written as it was found).

Use hyphen in all manuscript, between numbers.

Author Response

Dear  reviewer, 

We are grateful to you for the valuable suggestions provided.

We put the revised manuscript (highlighted in green) in attachment file.

Response to Reviewer 1 Comments

Comments and Suggestions for Authors

Overall, the paper is interesting and was consistently improved. A few short suggestions for authors:

Introduction

Line 36:  10–20%

Response: Thanks for pointing this out. We have used the hyphen format between numbers in revised manuscript.

Lines 42-43: Hyperparasitic fungi have been reported for various plant pathogenic fungi [9–11], and some of them have been used to control plant diseases [11–13].

Response: Thank you for your valuable suggestion. We have re-written this sentence according to your advicein revised manuscript.

Materials and methods

Consider not using the 1st person when describing your work (for example, we found can be written as it was found).

Response: Thank you for your valuable suggestion. We have re-written this part as “Wheat cultivar “Fielder” inoculated with Pst urediospores were kept in a growth chamber at about 16℃ and 80-90% relative humidity. When Pst was sporulating 14 days after inoculation, Pst uredinia started to change color from yellow to taupe.”

Use hyphen in all manuscript, between numbers.

Response: Thanks for pointing this out. We have corrected the hyphen format between numbers in revised manuscript.

Reviewer 2 Report

Thanks to the authors for taking into account the different comments in order to improve the quality of this article. Nevertheless, a few improvements still need to be made to this manuscript. I would also strongly recommend to the authors to take the time to check the additions made (carefully!) as some errors are found in it (part mat et meth!).

Major revisions :

Introduction part :

L46-47 : “In field experiments, …. Black scurf of potato was evaluated”.  Please rephrase the sentence by giving more scientific information and not just quoting the trial and "was evaluated": (1) at least the protection rate obtained or a % of reduction of symptoms (2) reminded that in this article the authors seem to highlight not only the mycoparasitic activity of the fungus but also the inducing effect on the plant defenses... So please take some precautions in our citation...

Materials and Methods

L119-L121 : Authors give the nucleotides sequences of the PST-EF1 and the wheat –EF1… but we find two probes couple of Pst-EF1….. thanks to correct !

L124 : “hyperparasite” urediospore or “hyperparasited” urediospore ? It’s not the same meaning !

Results :

L180 : “fungal signs”… what are fungal signs, thank to reformulate this sentence and  be more accurate !

L228-229 : Added sentence, very redundant to the sentence L226-228. I don’t well understand what is the idea develop by this sentence… maybe reformulate in a simple phrase to recall the the principle of action of hyperparasitic fungi.

Minor revisions

L47: erase th “s” at “a parasites fungi”

L58 : C. cladosporioides and not Cladosporium cladosporioides, authors have already cited the “complete” name at l50

L71 : Add “C. cladosporioides” before R23Bo

L169 : Legend of figure 3 : change “charcateristics” to “characteristics”

L204 suspension and not susponsion

Figure 6 : in the figure, authors have indicated CYR31+stain R23Bo, thank to add a “r” at strain !

L210: in the legend of the figure 6, add for the two times where authors cited the parasitic fungi the strain reference of C. cladosporioides that we observed in the figure : “Comparison of the yellow-colored …. and time point treatments of Cladosporium cladosporioides R23Bo”. But used C. cladosporioides L213 and L215. And use the C. cladosporioides and not Cladosporium cladosporioides.

L236, 257 : change conodian to conidian

Author Response

Dear  reviewer, 

We are grateful to you for the valuable suggestions provided.

We put the revised manuscript (highlighted in green) in attachment file.

Response to Reviewer 2 Comments

Major revisions :

Introduction part :

L46-47 : “In field experiments, …. Black scurf of potato was evaluated”.  Please rephrase the sentence by giving more scientific information and not just quoting the trial and "was evaluated": (1) at least the protection rate obtained or a % of reduction of symptoms (2) reminded that in this article the authors seem to highlight not only the mycoparasitic activity of the fungus but also the inducing effect on the plant defenses... So please take some precautions in our citation...

Response: Thank you for your valuable suggestion. We have re-written this sentence as “The cell wall protein fraction of Pythium oligandrum, a parasite of pathogen Rhizoctonia solani AG-3, enhanced the expression of defense-related genes in potato. Application of P. oligandrum oospore suspension at 104  or 105 mL−1 to potato seed tuber pieces, the black scurf disease severity index reduced about 35% compared with untreated controls in field”.

Materials and Methods

L119-L121 : Authors give the nucleotides sequences of the PST-EF1 and the wheat –EF1… but we find two probes couple of Pst-EF1….. thanks to correct !

Response: Thanks for pointing this out and we apologize for our oversight. The first probe is wheat-EF1 and we have corrected this mistake in revised manuscript.

L124 : “hyperparasite” urediospore or “hyperparasited” urediospore ? It’s not the same meaning !

Response: Thanks for pointing this out and we apologize for our inaccurate description. We have corrected as “hyperparasited urediospores” in revised manuscript.

Results :

L180 : “fungal signs”… what are fungal signs, thank to reformulate this sentence and  be more accurate !

Response: Thanks for pointing this out and we apologize for our inaccurate description. We have re-written this sentence as “Wheat leaves inoculated only with the conidian suspension of isolate R23Bo did not show any symptoms or signs of fungal infection”.

L228-229 : Added sentence, very redundant to the sentence L226-228. I don’t well understand what is the idea develop by this sentence… maybe reformulate in a simple phrase to recall the the principle of action of hyperparasitic fungi.

Response: Thanks for pointing this out. We planned to stress the significance of hyperparasitic fungi. However, due to the redundancy of the sentence L228-229, we have deleted it in revised manuscript.

Minor revisions

L47: erase th “s” at “a parasites fungi”

Response: Thanks for pointing this out. We have corrected it in revised manuscript.

L58 : C. cladosporioides and not Cladosporium cladosporioides, authors have already cited the “complete” name at l50

Response: Thanks for pointing this out. We have corrected it in revised manuscript.

L71 : Add “C. cladosporioides” before R23Bo

Response: Thanks for pointing this out. We have added “C.cladosporioides before R23Bo.

L169 : Legend of figure 3 : change “charcateristics” to “characteristics”

Response: Thanks for pointing this out. We have corrected it in revised manuscript.

L204 suspension and not susponsion

Response: Thanks for pointing this out. We have corrected this word in revised manuscript.

Figure 6 : in the figure, authors have indicated CYR31+stain R23Bo, thank to add a “r” at strain !

Response: Thanks for pointing this out and we apologize for our oversight.We have corrected this word in Figure 6.

L210: in the legend of the figure 6, add for the two times where authors cited the parasitic fungi the strain reference of C. cladosporioides that we observed in the figure : “Comparison of the yellow-colored …. and time point treatments of Cladosporium cladosporioides R23Bo”. But used C. cladosporioides L213 and L215. And use the C. cladosporioides and not Cladosporium cladosporioides.

Response: Thanks for pointing this out and we apologize for our inaccurate description. We have unified the description as “C. cladosporioides R23Bo” in the legend of the figure 6.

L236, 257 : change conodian to conidian

Response: Thanks for pointing this out. We have corrected this word in revised manuscript.

This manuscript is a resubmission of an earlier submission. The following is a list of the peer review reports and author responses from that submission.

Round 1

Reviewer 1 Report

Abstract

Lines 14, 24, 28 and elsewhere in the paper: use italic for Pst.

Response: Thanks for pointing this out. We have fixed this word in revised manuscript.

Line 16: identified as

Response: Thanks for pointing this out. We have deleted word “it” in revised manuscript.

This section needs to be improved, more emphasis can be placed on the practical applicability of Cladosporium cladosporioides.

Response: Thank you for your valuable suggestion. We have re-written this part as “Wheat rust outbreaks have caused significantly economic losses all over  the world. Puccinia striiformis f. sp. tritici (Pst) is an obligate biotrophic fungus causing stripe rust on wheat. Application of fungicides may cause environmental problems. The effects of hyperparasites on plant pathogens are the basis for biological control of plant pathogenic fungi and parasites of Pst have great value in biological agents development. Here, we report the isolation and characterization of isolate of Cladosporium cladosporioides from Pst based on morphological characterization and analysis of molecular markers. The hyperparasitic isolate was isolated from taupe-colored uredinia of Pst. Upon artificial inoculation, the hyperparasitic isolate was able to reduce the production and germination rate of Pst urediospores, and Pst uredinia changed color from yellow to taupe. Scanning electron microscopy demonstrated that the strain could efficiently colonize Pst urediospores. Therefore, the isolate has the potential to be developed into a biological control agent for managing wheat stripe rust”.

Keywords

In this section it is preferable not to repeat words from the title, but to emphasize other suggestive words regarding the content of the paper.

Response: Thank you for your valuable suggestion. We have re-written Keywords as “Puccinia striiformis f. sp. tritici (Pst), Cladosporium cladosporioides, hyperparasite, Microbial biological control agents (MBCAs)”.

Be sure to use italics for microorganism names throughout the entire paper.

Response: Thanks for pointing this out. We have fixed this mistake in revised manuscript.

Introduction

The sentences that make up this section sound like telegrams, are extremely short and have no scientific character. In fact, the whole section is extremely short and does not provide enough information on the subject approached by the authors.

In addition, the structure of the section is completely wrong. The authors must describe the subject approached, from the point of view of the current state of the literature, and only finally declare the purpose of their own research and the practical importance of their own results or observations, the contribution they bring in the field. It is not at all logical to say in the first paragraph what they discovered and then to refer to the literature and then to return to their own discoveries.

Therefore, I believe that this section needs to be rewritten and significantly improved in order to correspond to an scientific paper.

Response: Thank you for your valuable suggestion. We have rewritten Introduction with your suggestions in revised manuscript.

Materials and methods

This section needs to be improved. The authors must describe as fully as possible the working method and the equipment or methods of analysis, so that a researcher or even a student in the field can replicate the experiments.

Response: Thanks for pointing this out. We have rewritten Materials and Methods part with your suggestions in revised manuscript.

Results / Discussion / Conclusion

This sections frame the paper as preliminary research study.

Recommendations: Both Introduction and Materials and Methods lack the scientific expression and soundness and a rigorous rewriting is required. In my opinion, in the present form, the paper does not qualify for publishing. Therefore, I recommend at least major revisions and checking the correctness of the translation and some aspects of English grammar.

Response: Thank you for your valuable suggestion. We have rewritten both Introduction and Materials and Methods part with your suggestions in revised manuscript.

Reviewer 2 Report

This study aims to isolate new potential agents that could help to control yellow rust of wheat, a cryptogamic disease that is distributed worldwide and whose negative impact on wheat yields classifies it as a disease of concern. This work is very interesting and the methods used are adequate. Nevertheless, an effort must be made at the level of the introduction but especially at the level of the discussion which is not completed. I strongly encourage the authors to rewrite this last part in order to bring to this work the value it deserves!

Response: Thank you for your valuable suggestion. We have rewritten both Introduction and discussion with your suggestions in revised manuscript.

Introduction :

I do appreciate relatively short introductions that focus quickly on the subject of the study,but here it is perhaps a bit too "short cut" and we don't really get to see the importance of this study for the protection of wheat against rust and the contribution of this work.

Response: Thank you for your valuable suggestion. We have rewritten Introduction with your suggestions in revised manuscript.

L34-35 : Authors explained that yellow rust “poses a great threat to wheat production worldwide”. Is it possible to have a idea about that, for exemple the yield reduction ? Quantitative and qualitative negative impacts on wheat ? what about the fight against this disease ? Some problems with pesticides ? Emergence of pesticides resistance in Pst populations ?

Response: Thank you for your comment.

For the danger of Pst, we have rewritten and add some sentences as “In 2000, 9 of the 64 major wheat producing countries reported severe losses in wheat yields caused by stripe rust [2]. In 2000-2012, about 88% of the world's wheat-producing areas were affected by stripe rust [3]. In China, the disease can reduce the yield of wheat by 10%-20%, and even more than 60% in extremely severe epidemic years [4,5]”.

For the pesticides, we have added sentences as “Wheat stripe rust is currently controlled mainly by developing resistant cultivars and applying pesticides when needed. Race-specific resistance is often circumvented by new races of the pathogen, and pesticides application adds extra cost and may adversely affect the environment [6-8]. Therefore, other strategies such as biological control could provide additional options for managing stripe rust”.

L46-48 : Authors explained that “several hyperparasitic fungi have been previously reported for Pst” such as Alternaria, Cladosporium , Lecanicillium Lecanii, Microdochium Typhula secies… What is the novelty of this study ?

Response: Thank you for your comment.

We have rewritten the sentence as “However, only six species, Alternaria alternata, Cladosporium cladosporioides, Lecanicillium Lecanii, Microdochium nivale, Simplicillium obclavatum and Typhula idahoensis have been reported to infect uredinia and urediospores of Pst [18, 21-23]”.

Although six species of hyperparasitic fungi of Pst have been reported previously, the overall number is still small. There is still a lot of unknown species that need to be isolated and further studied. More hyperparasitic fungi of Pst could provide additional biological control options for managing stripe rust. Furthermore, characterization of newly isolated hyperparasitic fungi can lead to a better understanding of the diversity of hyperparasites, and will lead to the discoveries of novel fungal species and the development of novel biocontrol agents

Redondant sentences :

L43 : Some of hyperparasitic fungi have been used in control of plant diseases [4-6]

L49 : Hyperparasitc fungi may have potential in control of plant pathogenic fungi.

Response: Thank you for your comment. We have deleted the redondant sentence “Hyperparasitc fungi may have potential in control of plant pathogenic fungi.”

“Hyperparasitc fungi may have potential in control of plant pathogenic fungi.” Can the authors argue a bit for the use of hyperparasitic fungi by giving some examples of biocontrol products marketed (hyperparasitic fungi) against plant diseases ? Not necessarily wheat rust, if there is none...

Response: Thank you for your valuable suggestion. We have added two examples as “Ampelomyces mycoparasites suppressed the sporulation rate of powdery mildew, and the infected plants regained vigour after Ampelomyces killed the pathogens [14]. In field experiments, after treatment of potato seed tubers with Pythium oligandrum , a parasites of Rhizoctonia solani AG-3, black scurf of potato was evaluated [15]” in revised manuscript.

Material and Method :

The idea of the material and method part of an article would be to allow an interested person to repeat the experiments carried out with the help of this part... but here it seems to me impossible since no data is given about the infection of plants by Pst for the pathogenicity tests. What is the origin of the urediniospores of Pst race CYR31 used for wheat inoculation ? What is the concentration of urediniospores used for infection ? Are they put in solution in water? autoclaved water? a buffer? Are they sprayed on the leaves or applied with a brush?

Response: Thank you for your comment. We have added a detailed description of Pst inoculation in revised manuscript.

The inoculations of Pst were done following the previously described methods [21]. Briefly, wheat plants (cv. Fielder) grown in a greenhouse for 20 days were first inoculated with urediospores of Pst race CYR31 collected from Su11 wheat. The collected urediospores of Pst race CYR31 were diluted with water to 20 mg·mL-1 and inoculated by brush. The Pst-inoculated plants were incubated in a dew chamber at 12℃ in dark for 24 h, and then grown in a growth chamber at 16℃ with 16 h light photoperiod.

L94 : “The inoculations were done following the previously described methods [12].” Inoculation of the hyperparasitic fungi ? or Pst ? Please be very precise in your sentences because two fungi are inoculated one after the other, so be careful to explain which fungus the authors are talking about.

Response: Thanks for pointing this out and we apologize for our inaccurate description. We have corrected and moved the sentence “The inoculations of Pst were done following the previously described methods [21]” to the opening of this paragraph. And we further described this inoculation method.

Thank you for making the paragraph concerning the molecular characterization and the phylogenetic anallyse appear one after the other, I don't understand why the paragraph presenting the tests of pathogenesis and hyperparasitism is found between these two ones.

Response: Thank you for your valuable suggestion. We have changed the order of the two methods to make it more logical in revised manuscript.

Results

L111-113 : “Observations using a scanning electronic microscope (SEM) showed that without hyperparasite infection, Pst uredinia were yellow-colored, had a normal shape and structure (Fig. 1 A)” : I didn't know that you could see the "yellow" color of the uredinia by using the SEM... please rewrite the sentence differently ! Remembering that the yellow color is previously seen by an optical microscope observation or by eye view.

Response: Thanks for pointing this out and we apologize for our oversight. The yellow color of Pst was observed by eye view previously. We have corrected this sentence as “Observations using a scanning electronic microscope (SEM) showed that without hyperparasite infection, Pst uredinia had a normal shape and structure (Fig. 1 A)”.

L113-114 : “When Pst uredinia appeared taupe-colored, the urediniospores were shriveled 113 at the early infection stage of the hyperparasite (Fig. 1 B-D)” Same remark, rewrite with more precision the sentence

Response: Thanks for pointing this out and we apologize for our oversight. We have corrected this sentence as “The Pst urediospores were shriveled at the early infection stage of the hyperparasite (Fig. 1 B-D)”.

Figures :

Figures 1, 3 and 7 : The photographs are particularly qualitative and clear, nevertheless in order to allow all to understand well your results, thank you to add on the photographs of the arrows presenting the elements explained: the urediniospore, the spores of the hyperparasitic fungi, germ tube, conidiophores... etc... especially for people who are not used to working with hyperparasitic fungi.

Response: Thank you for your valuable suggestion. We have added indicating arrows in figures.

Figure 4 : The tree is not very qualitative, is it possible to increase the quality of the figure ?

Response: Thank you for your valuable suggestion. Your suggests we have considered before. We have notice that R23Bo, the hyperparasitic fungi we isolated, has a close relation with a published Pst hyperparasitic fungi, 14PI001. In order to better illustrate the species evolutionary relationship of R23Bo with other fungus, especially with 14PI001, we adjusted the layout of the evolutionary tree.

Figure 6 : This compound figure is very interesting but difficult to understand, is-it possible to more explained the various parameters following by description in the materiel and method ? For example, how the fertility of spores is estimated/ controlled ?

Response: Thank you for your valuable suggestion. The fertility of spores is estimated by germination rate of Pst urediospores. The method have been added in the Material and Method section.

For the results of the figure 6B, thank to authors to explain why there are only the kinetic point 3 and 5 hours … If the fertility of the urediniospiores is significantly affected at times 3 and 5 dpi, why is there no significant effect on the amount of DNA at these times? and the number of uredinium/leaves at 3 dpi?

Response: Thank you for your comment. The fertility of the urediniospiores were estimated by germination rate of Pst urediospores in vitro. The fungal biomass was measured according the Pst DNA content/wheat DNA content as the standard cave of Pst DNA and wheat DNA. The fungal biomass and the number of uredinium/leaves were dependent onPst growth and development  in vivo.

L177-178 : “the number of yellow uredinia per cm2 leaves was the lowest in the treatment of C. cladosporioides 3 days after Pst inoculation (Fig. 6A)” Are you sure ? For me it was at 9 days after infection of the plant by Pst, or I misunderstood figure 6A... so maybe review this figure...

Response: Thanks for pointing this out and we apologize for our oversight.We have corrected this sentence as “the number of yellow uredinia per cm2 leaves was the lowest in the treatment of C. cladosporioides 9 days after Pst inoculation”.

Discussion :

L 230- 252 : To be moved to the introduction, not its place in the discussion. An effort must be put into the writing of the discussion which lacks subtlety. It is your results that need to be discussed!

Response: Thanks for pointing this out. We have moved these sentences to the introduction, and rewritten Introduction and Discussion with your suggestions in revised manuscript.

General comments or minor modifications :

We sometimes find the spelling urediniospores and urediospores in the text or in the figures, both are correct, but is it possible to choose one of the two terms and then use only one ? To be more homogeneous in our study ?

Response: Thanks for pointing this out. We have unified the word and changed as “urediospores” in revised manuscript.

L200 : “the urediospores” add a “space” between “the” and “urediniospores”

Response: Thanks for pointing this out.We have corrcted this mistake in revised manuscript.

L211: “potential to”, checked the size of the space between this two words.

Response: We have checked the size of this two words in a right space .